# Does the Implementation of a Quality Improvement Care Bundle Reduce the Incidence of Acute Kidney Injury in Patients Undergoing Emergency Laparotomy?

**DOI:** 10.3390/jcm8081265

**Published:** 2019-08-20

**Authors:** James F. Doyle, Alexander Sarnowski, Farzad Saadat, Theophilus L. Samuels, Sam Huddart, Nial Quiney, Matthew C. Dickinson, Bruce McCormick, Robert deBrunner, Jeremy Preece, Michael Swart, Carol J. Peden, Sarah Richards, Lui G. Forni

**Affiliations:** 1Department of Intensive Care Medicine and Surrey Peri-Operative Anaesthesia and Critical Care Collaborative Research Group (SPACER), Royal Surrey County Hospital NHS Foundation Trust, Guildford, Surrey GU2 7XX, UK; 2Department of Intensive Care Medicine, Royal Brompton & Harefield NHS Foundation Trust, London SW3 6NP, UK; 3Department of Anaesthesia and Intensive Care Medicine, Surrey & Sussex Healthcare NHS Trust, Redhill RH1 5RH, UK; 4Department of Anaesthesia, Royal Devon & Exeter NHS Foundation Trust, Exeter EX2 5DW, UK; 5Department of Anaesthesia, Torbay & South Devon NHS Foundation Trust, Torquay TQ2 7AA, UK; 6Department of Anaesthesia, Royal United Hospitals Bath NHS Foundation Trust, Avon BA1 3NG, UK; 7Department of Surgery, Royal United Hospitals Bath NHS Foundation Trust, Avon BA1 3NG, UK; 8Department of Clinical & Experimental Medicine, Faculty of Health and Medical Sciences, University of Surrey, Guildford Guildford, GU2 7YS, UK

**Keywords:** post-operative complications, acute kidney injury, enhanced recovery, goal directed therapy, emergency surgery, laparotomy

## Abstract

Purpose: Previous work has demonstrated a survival improvement following the introduction of an enhanced recovery protocol in patients undergoing emergency laparotomy (the emergency laparotomy pathway quality improvement care (ELPQuiC) bundle). Implementation of this bundle increased the use of intra-operative goal directed fluid therapy and ICU admission, both evidence-based strategies recommended to improve kidney outcomes. The aim of this study was to determine if the observed mortality benefit could be explained by a difference in the incidence of AKI pre- and post-implementation of the protocol. Method: The primary outcome was the incidence of AKI in the pre- and post-ELPQuiC bundle patient population in four acute trusts in the United Kingdom. Secondary outcomes included the KDIGO stage specific incidence of AKI. Serum creatinine values were obtained retrospectively at baseline, in the post-operative period and the maximum recorded creatinine between day 1 and day 30 were obtained. Results: A total of 303 patients pre-ELPQuiC bundle and 426 patients post-ELPQuiC bundle implementation were identified across the four centres. The overall AKI incidence was 18.4% in the pre-bundle group versus 19.8% in the post bundle group *p* = 0.653. No significant differences were observed between the groups. Conclusions: Despite this multi-centre cohort study demonstrating an overall survival benefit, implementation of the quality improvement care bundle did not affect the incidence of AKI.

## 1. Introduction

Enhanced recovery pathways are now integral in many surgical pathways in order to optimize patient care, with the aim of reducing both post-operative morbidity and mortality [1]. The application of standardized pathways has been shown to reduce both post-operative complications and length of stay in elective surgery [2]. The adoption of care bundles in order to improve outcomes has been applied to both scheduled non-emergent surgery and also to emergency surgery. Recently published data reported a significant case mix-adjusted risk of death reduction from 15.6% to 9.6% at 30 days following the implementation of the emergency laparotomy pathway quality improvement care (ELPQuiC) bundle [3]. This pathway is comprised of the following steps:All emergency surgical admissions risk assessed using the M(EWS) score [4]. Those with M(EWS) ≥4 reviewed by critical care outreach team.Broad spectrum antibiotics given to all patients with suspicion of peritoneal soiling or with sepsis.Once the decision is made for a laparotomy then next available theatre slot is used (or within 6 h) with senior clinical input (consultant anaesthetist and surgeon).Resuscitation commenced using goal-directed techniques and continued for a minimum of six hours post-operatively.All patients admitted to critical care when possible after surgery or held in a post anaesthetic care unit for at least six hours.

The need for such an integrated approach was highlighted in the UK when the Emergency Laparotomy Network group published data on 1800 patients showing a 30 day mortality of 14.9%, which rose to 24.4% in patients aged 80 and over [5]. This high mortality was also demonstrated in other countries with differing healthcare systems [6,7]. Following the evidence of such high mortality, standards of care were developed in the UK which recommended defined pathways with evidence-based interventions for all high risk and emergency surgical patients [8]. The use of a care-bundle concept is not new in critical care with several successful examples in current practice, such as the Surviving Sepsis Campaign with substantial morbidity and mortality improvements observed through the global implementation of this care bundle [9].

The observed mortality and morbidity in high risk groups of patients admitted to intensive care, including patients undergoing emergency surgery, remains significant with acute kidney injury (AKI) being a major factor complicating critical illness. AKI is associated with a mortality rate of up to 60% [10]. The relevance of AKI in emergency surgery is reflected in the results from the AKI EPI study where AKI complicated 51% of elective surgical patients admitted to ICU, and increased further to 56% in those undergoing emergency surgery [11]. This is further compounded in elderly patients with higher rates of AKI and worse mortality [12]. Furthermore, the long term sequelae after an episode of AKI are substantial, with a single episode of AKI independently associated with an increase in 10-year mortality [13]. Currently available treatment strategies have the potential to improve patient outcome and provide considerable health savings if implemented early [14]. The implementation of the ELPQuiC bundle was associated with significant increase in the use of goal directed fluid therapy and admission to ICU across the participating sites. These interventions form part of the KDIGO (Kidney Disease Improving Global Outcomes) clinical practice guidelines for the management of AKI, which recommend maintenance of perfusion pressure, functional haemodynamic monitoring and ICU admission [15]. It is unclear whether the adoption of such a goal directed approach in high risk patients may result in a reduction in AKI or indeed whether the development of AKI in this group is specifically associated with worse outcomes [16]. Given the observed risk-adjusted mortality improvement seen in the ELPQuiC study, we examined the data to see if this effect could be explained by a reduction in AKI translating into a survival benefit.

## 2. Methods

Development and components of the care bundle are described elsewhere [3,17]. The ELPQUiC study was conducted in 4 acute hospital trusts in the United Kingdom, with an intervention period from December 2012 to July 2013 after a baseline monitoring period. A multi-centre cohort subgroup analysis was performed with data gathered from the original ELPQuiC study. Colleagues in the ELPQuiC collaborator group accessed the relevant components of their ELPQuiC raw data. Where needed, additional biochemical data was obtained from the hospital’s electronic pathology system. All data was reviewed by a second investigator.

AKI was defined as described by the KDIGO serum creatinine thresholds only. Urine output thresholds were not used, as data for this was not complete. We defined the reference or baseline creatinine as the lowest preoperative serum creatinine from the 12 months prior to admission. Serum creatinine values at baseline, immediately post-operatively (within the first 24 h but usually within hours of surgery completion), on day 30 and the maximum recorded creatinine between day 1 and day 30 were taken. P-POSSUM (Physiological and Operative Severity Score for the enumeration of Mortality and morbidity) and 30-day mortality data were also collected [18]. CKD stage was identified via the MDRD (Modification of Diet in Renal Disease study) equation with age, gender and baseline creatinine [19].

The primary outcome was the incidence of AKI in the pre- and post-ELPQuiC bundle patient population. Secondary outcomes included the KDIGO stage specific incidence of AKI.

As this project was an assessment of current practice and implementation of best-practice guidelines, it was confirmed by the National Research Ethics service that formal ethical approval was not required [3].

### Statistical Analysis

For discrete data, we used Pearson’s chi-squared test, Fisher’s exact test, Wilcoxon rank sum test and the Mantel-Haenszel odds ratio (stratifying for centre), along with the φ-coefficient or Goodman-Kruskal γ statistic where appropriate. In addition, we used four-fold plots to provide a visual representation of the odds ratios, which align the vertical and horizontal quadrants with an odds ratio equal to 1. This also permits the use of confidence rings that provide a visual indication for the test of no association; they will only overlap if and only if the observed counts are consistent with the null hypothesis. Furthermore, the width of the confidence rings provides a visual guide to the precision of the data. All analyses were carried using the open source statistical package R (Foundation for Statistical Computing, Vienna, Austria) [20], along with ggplot2 [21], Forest plot [22] and vcd packages [23].

## 3. Results

Table 1 shows the baseline demographics and outcomes obtained from the four centres. A total of 292 patients pre-ELPQuiC bundle and 424 patients post-ELPQuiC were identified across the four centres with no significant differences observed between the groups. Ten patients from the initial ELPQuiC study were excluded due to incomplete data on renal function. There was no significant difference in P-POSSUM scores pre- or post-ELPQuiC implementation: the pre-ELPQuiC median was 9.0% (IQR 2.9–27.0%) versus the post-ELPQuiC median of 8.6% (IQR 3.5–31.4%) (Wilcoxon rank sum test *p* = 0.5842; Figure 1). However, although the baseline CKD rates in the pooled post-ELPQuiC group were significantly higher than in the pre-ELPQuiC group, *p* = 0.01961, the Goodman and Kruskal γ statistic of 0.036 suggests this is a very weak association. Moreover, this is for all CKD, if one considers only CKD stages 3 to 5 (the highest risk of AKI) then there is no difference (*p* = 0.19).

### 3.1. Day 1 AKI

For our primary outcome; incidence of AKI between pre- and post-ELPQuiC implementation on day 1 post-op, the Mantel-Haenszel odds ratio (95% CI) for the four centres was 0.93 (0.72, 1.61). Using four-fold plots, crude numbers for day 1 incidence of AKI for each centre demonstrate that the odds ratios do not significantly differ from 1, but that the precision of the data is low (Figure 1 and Figure 2). Additionally, the cumulative rates of AKI for day 1 post-surgery were 18.4% versus 19.8% (pre- and post-pathway, respectively), with the data for each centre and combined data for all 4 centres showing no statistical significance (Centre 1; *p* = 0.460, Centre 2; *p* = 0.346, Centre 3; *p* = 0.319, Centre 4; *p* = 0.817, pooled *p* = 0.686).

There was no significant association between the incidence of KDIGO defined AKI and the use of the pathway on the first post-operative day for each individual centre or when the data was merged.

### 3.2. Maximum AKI Day 1–30

Using the maximum creatinine level and associated AKI incidence between days 1 and 30 post-surgery, the Mantel-Haenszel odds ratio (95% CI) for the 4 centres was 0.87 (0.61, 1.24) (Figure 3). There was no significant association between the incidence of AKI and the use of the pathway (Centre 1; *p* = 0.137, Centre 2; *p* = 0.501, Centre 3; *p* = 0.388, Centre 4; *p* = 0.680) or when the data was pooled (*p* = 0.740) (Figure 4B). For the maximum creatinine levels (when assessed using KDIGO criteria) there was no significant association demonstrated for each individual centre or when the data was merged.

### 3.3. Day 30 AKI

On day 30 after surgery, the Mantel-Haenszel odds ratio (95% CI) for the 4 centres was 0.56 (0.31, 1.00) (Figure 3). Using four-fold plots, crude numbers demonstrate that the day 30 incidence of AKI does not significantly differ from day 1, but the precision of the data is very low and varied across the centres (Figure 4).

There was no significant association between the incidence of AKI and the implementation of ELPQuiC at day 30 post-operatively for each individual centre (Centre 1 *p* = 1.00, Centre 2 *p* = 1.00, Centre 3 *p* = 0.077, Centre 4 *p* = 0.241). However, a small and weak association was observed when the data was pooled (Figure 4C) *p* = 0.069, phi-coefficient 0.09).

In a comparison of KDIGO AKI subgroups, again no significant difference was found on either day 1 (*p* = 0.5321) or day 30 (*p* = 0.1516) using crude data.

However, when correcting for rates of AKI for pre-existing CKD the Mantel-Haenzel Chi Squared test confirmed that the incidence of AKI relating to ELPQuiC implementation is not statistically associated with pre-existing CKD (Day 1 AKI *p* = 0.771, Max day 1–30 AKI *p* = 0.929, Day 30 AKI *p* = 0.087).

### 3.4. Mortality

Mortality incidence at 30 days was reported by the original ELPQuiC study and there was no significant association between the incidence of unadjusted 30-day mortality and the implementation of ELPQuiC for each individual centre (Figure 5 and Figure 6, respectively) or when the data was pooled (*p* = 0.08) (Figure 7), this is in keeping with the original ELPQuiC paper [3], which then identified the risk-adjusted survival benefit.

## 4. Discussion

Our results suggest that the implementation of a quality improvement care bundle, although conferring a survival benefit, does not affect the incidence of AKI in the immediate post-operative period or in the 30 days after surgery. This is true for both the individual institutions and the cumulative data from all four centres. Therefore, it is reasonable to conclude that the survival benefit seen in the original study is not related to a reduction in AKI, as defined by changes in serum creatinine. This is in keeping with studies on goal-directed therapy in surgery, where a benefit in terms of renal outcomes tends to be observed mainly where vasopressors are employed together with goal directed fluid therapy. The effects of goal-directed fluid therapy have been examined in major gastrointestinal surgery. The OPTIMISE trial randomised 734 high risk patients across 17 UK centres who were undergoing major gastrointestinal surgery to receive usual care or goal-directed fluid therapy and inotropy (dopexamine) to achieve stroke volume targets. Observer blinding was used. Primary outcomes included moderate or major complications and mortality. These outcomes occurred in 36.6% of patients in the intervention group and 43.4% in the usual care group giving a relative risk of 0.84 (95% CI 0.71–1.01; *p* = 0.07), which failed to reach statistical significance. Secondary outcomes, including infection, length of stay, mortality at 30 and 180 days, and morbidity at day 7 were also no different between the cohorts [24]. However, meta-analyses and systematic reviews have suggested some benefits of peri-operative goal directed therapy. A 2014 Cochrane review by Grocott et al. of 31 studies with 5292 patients showed reductions in renal failure, respiratory failure and wound infections in intervention groups. Fewer patients suffered complications in the intervention groups and on average their hospital length of stay was 1.16 days shorter. Mortality and total time in critical care was no different. However, the review was limited by a single large study that exerted a sizeable influence on the overall data pool and this must be considered when interpreting the results. It was noted that there seemed to be no harm associated with the use of goal-directed therapy; and therefore, with some putative benefits it may be a reasonable peri-operative strategy [25].

In the ELPQuiC study, commencement, dosing or timing of vasopressor use was not protocolised in either group. This may be of relevance given that vasopressin, for example, has achieved popularity as the vasopressor of choice in terms of limiting the degree of AKI, notably most recently in the VANISH trial for sepsis associated AKI [26]. It is unknown whether unit preferences for vasopressors affected the incidence of AKI in this study or affected the observed mortality benefit. Interestingly the incidence of overall AKI at day one post-surgery is about 50% of that reported in AKI-EPI, although this probably reflects the fact that all comers were admitted to ICU or held in a post-anaesthesia recovery area for an extended period, where the incidence of AKI would be expected to be less. Furthermore, given that 25% of patients had AKI at day 1 this almost certainly implies that the AKI was present prior to surgery given the creatinine kinetics following AKI. This gives further support to the importance of caring for these patients in a critical care setting where renal function and fluid balance can be closely monitored. The use of vasopressors may limit intra-operative hypotension (a risk factor for AKI), the extent and duration of which relate to the severity of the renal insult. Sun et al. conducted a retrospective analysis of 5127 patients undergoing elective non-cardiac surgery to delineate the relationship of intra-operative hypotension with renal outcomes. Mean arterial pressures of less than 60 mmHg for more than 20 min or less than 55 mmHg for more than 10 min were associated with increased risk of acute kidney injury [27]. This is unsurprising given the findings in both animal and human studies, which have demonstrated that renal blood flow becomes pressure-dependent when mean arterial pressure falls below the level at which the kidneys can autoregulate. Outside this window, renal blood flow declines with reductions in mean arterial pressure. Renal perfusion is also dependent on an adequate cardiac output in addition to a sufficient mean arterial pressure [28,29]. We did not stratify according to the presence of intra-operative hypotension, or additional risk factors for AKI, including use of nephrotoxic drugs, sepsis or volume depletion in the pre-operative period. These risk factors may be particularly relevant to the population undergoing emergency laparotomy. Moreover, we have defined AKI solely by serum creatinine, which is relevant in that several studies that have demonstrated a reduction in AKI post operatively with the use of a care bundle, principally observed an increase in urine output and hence a reduction in AKI using this criteria [30,31]. Given the heterogeneity in terms of AKI, it is unlikely that such an approach would influence AKI rates, but this provides further support for the observed mortality benefit being a product of global improvement in care rather than one aspect.

Consensus in terms of the nomenclature of AKI [32,33], has allowed a considerable body of evidence to accumulate regarding the epidemiology and pathophysiology of this syndrome. However, the methods of assessing renal function remain limited. For this study, creatinine was used as the renal biomarker, which has significant limitations outside steady state conditions [34]. Utilising more specific renal biomarkers may unmask “sub clinical AKI” and improved renal recovery with the implementation of a quality improvement care bundle, which may yet explain the survival improvement, particularly if sustained. Moreover, given the data on long term morbidity and mortality following an episode of AKI, any possibility of significantly reducing AKI after emergency laparotomy would be expected to see improved long term benefits.

Since the ELPQuIC project was published, the National Emergency Laparotomy Audit data has documented outcome data on almost 24,000 cases [35]. The 30 day mortality across all English and Welsh hospitals appears to have decreased from the 14.9% previously described in the US and UK [3,6] to 9.5%, according to the fourth NELA report on 2017 data. It is likely that the increased focus on patients undergoing emergency surgery and the quality improvement approach used in the ELPQuIC study and the larger EPOCH and Emergency Laparotomy Collaborative studies [36] are helping to improve outcomes in this high risk group of patients.

Limitations of this study include the lack of urinary output data for AKI classification and assessment of fluid balance. Using serum creatinine alone in such a heterogenous group may lead to inaccuracies in GFR estimation given changes in creatinine metabolism as well as the effects of administered drugs, although it seems unlikely that this was different in the two groups. Data was also lacking on the rates and duration of any renal replacement therapy (RRT) provided to those patients with an AKI 3. However, in the initial post-operative period the rate of RRT in one of the four centres was 22% amongst those patients classified as having AKI 3. Given a larger number of study participants it is possible that the strength of the association for ELPQuiC implementation and a reduction of AKI by day 30 would improve, however, the aim of this subgroup analysis was not to consider the power to detect AKI but rather, the reason for the identified risk-adjusted survival benefit.

## 5. Conclusions

This multi-centre cohort subgroup analysis suggests that the implementation of a quality improvement care bundle did not affect the incidence of AKI. This is in contrast to the survival benefit demonstrated using a care bundle and provides the stimulus to clarify the factors that may yet improve AKI in this high-risk patient group.

## Figures and Tables

**Figure 1 jcm-08-01265-f001:**
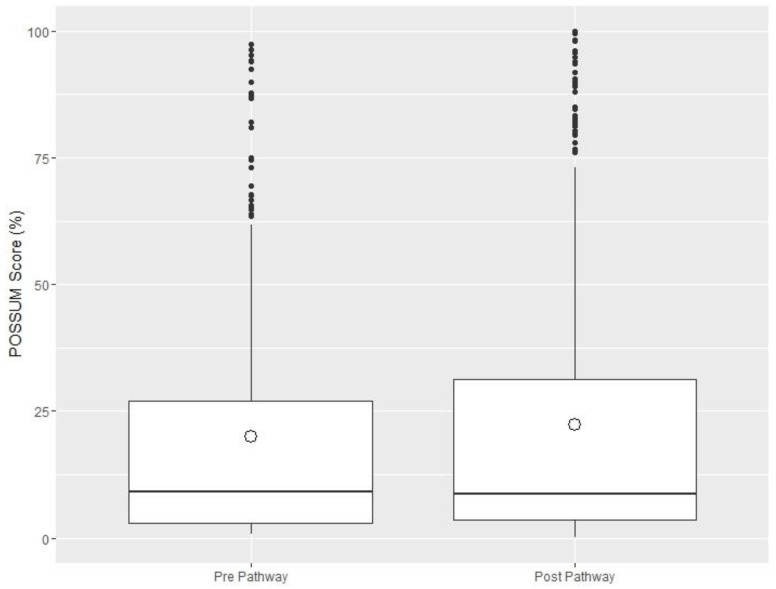
Cumulative P-POSSUM scores pre- and post-ELPQuiC implementation—circle represents the mean value.

**Figure 2 jcm-08-01265-f002:**
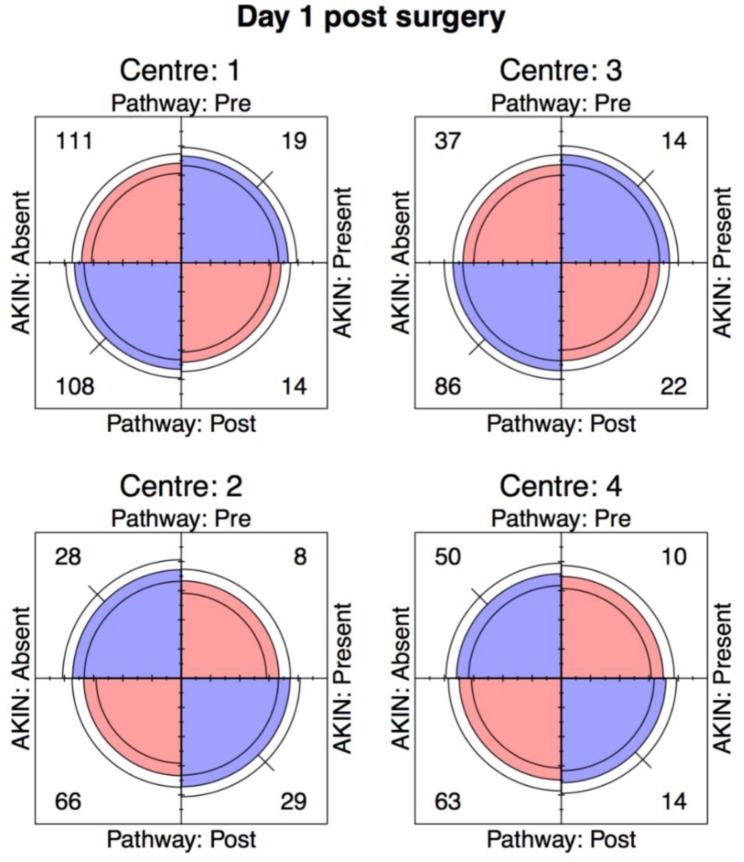
Four-fold Plot. Incidence of AKI day 1 post-op.

**Figure 3 jcm-08-01265-f003:**
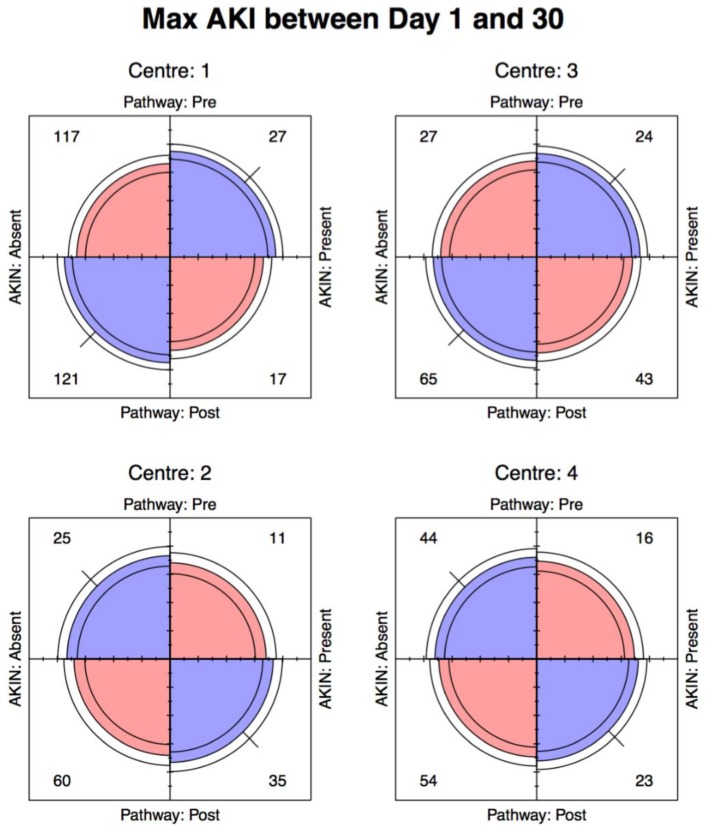
Four-fold Pl–t. Incidence of maximum AKI obtained between day 1 and day 30 post-op.

**Figure 4 jcm-08-01265-f004:**
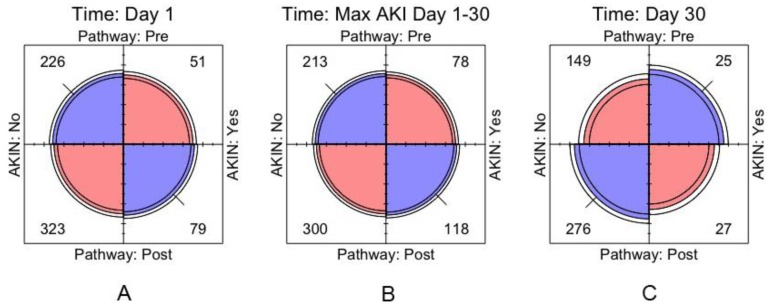
Cumulative AKI incidence (**A**) Day 1, (**B**) Max (day1-day30) and (**C**) Day 30 post-op pre and post-ELPQuiC implementation:

**Figure 5 jcm-08-01265-f005:**
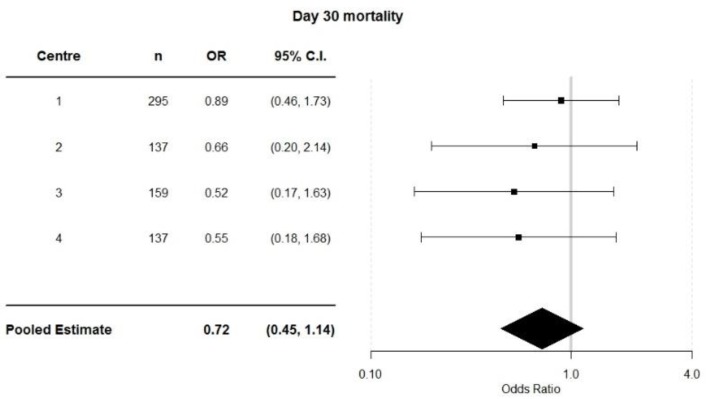
Incidence of 30-day mortality.

**Figure 6 jcm-08-01265-f006:**
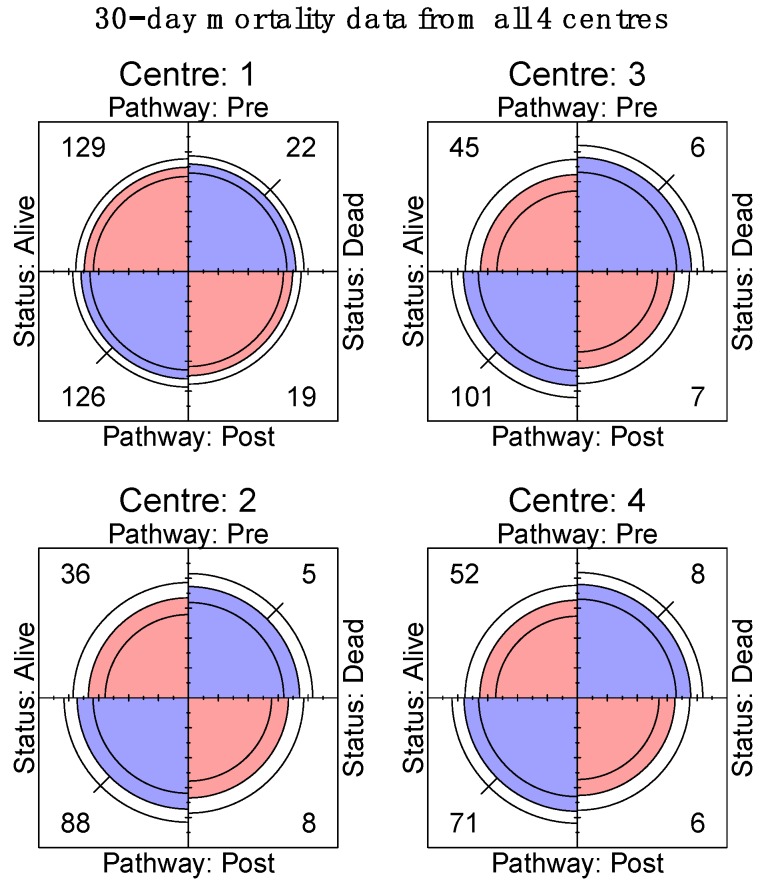
Centre specific 30-day mortality data.

**Figure 7 jcm-08-01265-f007:**
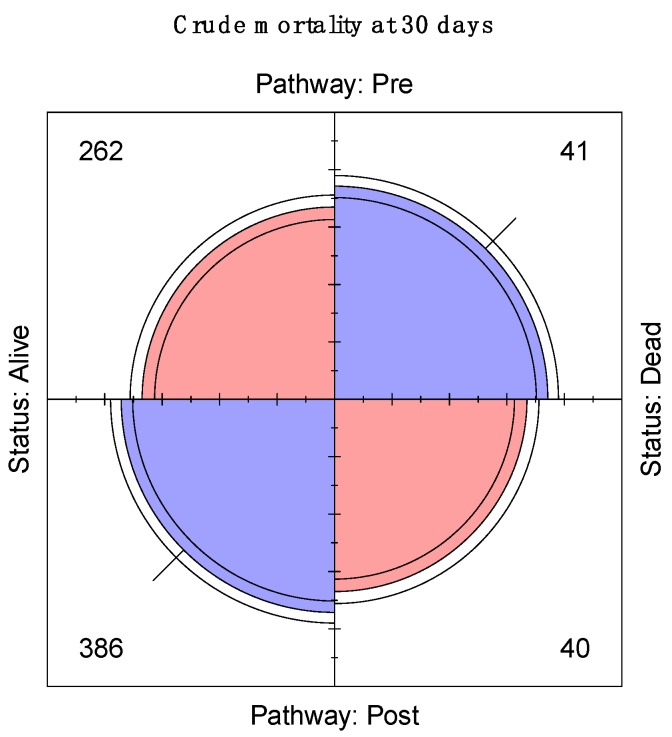
Cumulative 30-day mortality data.

**Table 1 jcm-08-01265-t001:** Demographics and outcomes of patients before and after inplementation of the emergency laparotomy pathway quality improvement care bundle.

	Site 1		Site 2		Site 3		Site 4		All Patients	
	Before ELPQuiC (*n* = 51)	After ELPQuiC (*n* = 109)	Before ELPQuiC (*n* = 144)	After ELPQuiC (*n* = 144)	Before ELPQuiC (*n* = 44)	After ELPQuiC (*n* = 97)	Before ELPQuiC (*n* = 60)	After ELPQuiC (*n* = 77)	Before ELPQuiC (*n* = 299)	After ELPQuiC (*n* = 427)
Age (years) *	66.6 (16.6)	65.3 (17.7)	65.1 (16.6)	63.7 (17.5)	65.7 (13.9)	69.3 (14.0)	66.2 (15.0)	66.0 (15.5)	65.6 (15.8)	65.8 (16.5)
Sex										
F	38 (75)	56 (51.4)	73 (50.7)	79 (54.9)	19 (43)	49 (51)	31 (52)	41 (53)	161 (53.8)	225 (52.7)
M	13 (25)	53 (48.6)	71 (49.3)	65 (45.1)	25 (57)	48 (49)	29 (48)	36 (47)	138 (46.2)	202 (47.3)
Outcomes at 30 days										
alive	42 (82)	96 (88.1)	123 (85.4)	126 (87.5)	39 (89)	89 (92)	53 (88)	71 (92)	257 (86.0)	382 (89.5)
dead	9 (18)	13 (11.9)	21 (14.6)	18 (12.5)	5 (11)	8 (8)	7 (12)	6 (8)	42 (14.0)	45 (10.5)
Died in hospital										
no	41 (80)	96 (88.1)	122 (84.7)	125 (86.8)	37 (84)	89 (92)	52 (87)	70 (91)	252 (84.3)	380 (89.0)
yes	10 (20)	13 (11.9)	22 (15.3)	19 (13.2)	7 (16)	8 (8)	8 (13)	7 (9)	47 (15.7)	47(11.0)
ASA fitness grade										
I	5 (10)	14 (12.8)	12 (8.3)	16 (11.1)	4 (9)	8 (8)	6 (10)	7 (9)	27 (9.0)	45 (10.5)
II	10 (20)	36 (33.0)	48 (33.3)	52 (36.1)	9 (21)	32 (33)	28 (47)	27 (35)	95 (31.8)	147 (34.4)
III	19 (37)	40 (36.7)	46 (31.9)	44 (30.6)	18 (41)	40 (41)	20 (33)	32 (42)	103 (34.5)	156 (36.5)
IV	16 (31)	18 (16.5)	31 (21.5)	26 (18.1)	12 (27)	12 (12)	5 (8)	10 (13)	64 (21.4)	66 (15.5)
V	1 (2)	1 (0.9)	7 (4.9)	6 (4.2)	1 (2)	5 (5)	1 (2)	1 (1)	10 (3.3)	13 (3.0)
Length of hospital stay (days) ^†^	11 (7–24)	11 (7–21)	12 (7–23)	10 (6–18)	12 (8–21)	12 (8–19)	10 (7–21)	13 (6–32)	11 (7–23)	11 (6–21)
P-POSSUM risk score *	0.226 (0.282)	0.251 (0.298)	0.193 (0.234)	0.267 (0.307)	0.200 (0.207)	0.179 (0.241)	0.179 (0.237)	0.159 (0.212)	0.197 (0.239)	0.223 (0.278)
*P* ^‡^	0.730		0.140		0.764		0.755		0.395	

Values in parentheses are percentages unless indicated otherwise; * values are mean (s.d.) and ^†^ median (i.q.r.) for survivors. ELPQuiC, emergency laparotomy pathway quality improvement care; ASA, American Society of Anesthesiologists; P-POSSUM, Portsmouth modification of Physiological and Operative Severity Score for the enumeration of Mortality and morbidity; ^‡^, test for proportions.

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
