# Peer review of "Does the Implementation of a Quality Improvement Care Bundle Reduce the Incidence of Acute Kidney Injury in Patients Undergoing Emergency Laparotomy?"

_jcm, 2019, doi:10.3390/jcm8081265_

Round 1
Reviewer 1 Report
Although the question whether mortality benefits from introduction of the ELPQuiC bundle are due to a decreased AKI incidence is relevant and considerable efforts have been made to answer this question, unfortunately the paper cannot answer this question since the baseline CKD rates are different between the two cohorts. It is well known that CKD is a risk factor for developing another AKI episode. Why this difference in the CKD rates has occurred can only be speculated. It is certainly quite unfortunate because the patient population shold not have changed pre and post introduction of the bundle. However, the original question cannot be answered based on these data.
Author Response
We thank the reviewer for this inciteful comment and apologise for any confusion. The CKD rates refer to all classes (CKD 0 to 5) but this is a weak association as evidenced by the gamma statistic. However, in terms of those at risk of AKI namely CKD 3 to 5 there was no difference and a sentence has been added to that end in the text outlined in red:
However, although the baseline CKD rates in the pooled post ELPQuiC group were significantly higher than in the pre ELPQuiC groups p=0.01961 the Goodman and Kruskal gstatistic of 0.036 suggests this is a very weak association. Moreover this is for all CKD if one considers just CKD stages 3 to 5 (the highest risk of AKI) then there is no difference (p=0.19).
We hope that this answers the reviewers comments.
Reviewer 2 Report
-Emergency laparotomy is a common surgical procedure undertaken for a wide variety of acute intra-abdominal conditions with high death rate reported until recently. Underlying this process successive analyses have found poor standards of care with wide variation in a number of key process indicators that are supported in evidence-based clinical guidelines. Implementation of the ELPQuiC bundle and demonstration of improved outcomes provides evidence of decreased mortality after emergency laparotomy. The introduction of a five-component care bundle led also to a significant reduction in P-POSSUM risk-adjusted 30-day mortality.
-If the observed mortality benefit could be explained by a difference in the incidence of pre vs post implementation AKI’s rate? The same positive effect observed by adoption of GDT and ICU postoperetive on overall mortality seems to be not transferable to the context of AKI. The overall AKI incidence was 18.4% in the pre-bundle group versus 19.8% in the post bundle group p=0.653.
-The overall incidence of AKI in ICU patients ranges from 20% to 50% with lower incidence seen in elective surgical patients and higher incidence in sepsis patients. Few studies describe postoperative AKI in noncardiac/nonvascular patients.
-Reported data are consistent with the submitted work’s results. Even if AKI represents a significant risk factor for mortality is not the only one involved in ICU patients. The risk factors for death of critically ill patients with AKI recognized in the literature include: older age, prolonged hospitalization, high APACHE II scores, comorbidities, oliguria, high lactate, hypovolemia, metabolic acidosis, sepsis, multiple trauma, vasoactive drugs, and need for invasive mechanical ventilation. Given the heterogeneity of risk factors and complex management of many of them it is likely that, in this huge multicenter but varied surgical population, ELPQuiC bundle has not been enough to change AKI incidence rate. Further studies are required to uncover limitations of relatively small sample examination and to clarify the potential influence of Enhanced Recovery programs (ELPQuiC bundle specific) on AKI incidence in emergent laparotomy.
Author Response
We thank the reviewer for these comments with which we completely agree. In particular the heterogeneity of causes in this cohort is appreciated therefore we have added to the manuscript the following in blue:
Given the heterogeneity in terms of AKI it may be unlikely that such an approach would influence AKI rates but this provides further support for the mortality benefit observed being a product of global improvement in care rather than one aspect.
Therefore we would stress that benefits are not from the effects on one system but a more holistic benefit. We do hope this address the reviewers comments regarding our paper.
Reviewer 3 Report
This is an interesting study presented by Doyle and colleagues about diagnostic & therapeutic strategy in patients undergoing emergency laparotomy. First evidence that this strategy may reduce significantly patients mortality was released in 2015
Huddart S1, Peden CJ, Swart M, McCormick B, Dickinson M, Mohammed MA, Quiney N; ELPQuiC Collaborator Group; ELPQuiC Collaborator Group.Br J Surg.Use of a pathway quality improvement care bundle to reduce mortality after emergency laparotomy. 2015 Jan;102(1):57-66. doi: 10.1002/bjs.9658 . Epub 2014 Nov 10.
However there were no data since now trying to compare achieved results.
Authors show on a quite large groups of patients that ELPQuiC bundle did not significantly increase the risk of AKI and patients mortality, both after 1 or 30 days after surgery. Interestingly, the risk of AKI was not correlated even in patients with chronic kidney disease history.
What is interesting, similar results that quality improvement programmes after emergency abdominal surgery may not increase patient survival were shown in a paper recently released.
Lancet. 2019 Apr 25. pii: S0140-6736(18)32521-2. doi: 10.1016/S0140-6736(18)32521-2 . [Epub ahead of print]
Effectiveness of a national quality improvement programme to improve survival after emergency abdominal surgery (EPOCH): a stepped-wedge cluster-randomised trial.
Peden CJ, Stephens T, Martin G, Kahan BC, Thomson A, Rivett K, Wells D, Richardson G, Kerry S, Bion J, Pearse RM; Enhanced Peri-Operative Care for High-risk patients (EPOCH) trial group.
This may point to future studies involving more patients and clarification of presented results.
As Authors mentioned, no analysis concerning patients urine output and catecholamines/vasopressors used may be a huge limitation of this study. Creatinine level measurement (with GFR estimation) may lead to misleading conclusions, when not compared to patients body mass, energetic status (kachexia) or type of food ingested and other drugs taken. Then urine output gives more information about patient's condition.
However, presented results about quality improvement strategy in patient's after emergency laparotomy does not increase the risk of AKI/death also show that this strategy is not harmful and give some hope for future studies.
Author Response
We thank the reviewer for the useful comments we agree regarding the limitations of serum creatinine alone in assessing AKI particularly in this cohort : to that end we have added a paragraph in green:
Using serum creatinine alone in such a heterogenous group may lead to inaccuracies in GFR estimation given changes in creatinine metabolism as well as the effects of administered drugs but it seems unlikely that this was different in the two groups.
Thank you for the feedback also : certainly there is some disparity in the literature and this may reflect the role of education and on-going audit in terms of health improvement projects whereby continual feedback results in better outcomes. Although interesting we feel that this is beyond the scope of this article as we were specifically examining the role (or not!) of AKI.
Round 2
Reviewer 1 Report
I would suggest moving the discussion of the higher baseline CKD rates in the post protocol implementation cohort from the results section to the discussion section.